# A Wireless Multi-Layered EMG/MMG/NIRS Sensor for Muscular Activity Evaluation

**DOI:** 10.3390/s23031539

**Published:** 2023-01-31

**Authors:** Akira Kimoto, Hiromu Fujiyama, Masanao Machida

**Affiliations:** Faculty of Science and Engineering, Saga University, 1 Honjo, Saga 840-8502, Japan

**Keywords:** electromyography, mechanomyography, near-infrared spectroscopy, layered sensor, wireless

## Abstract

A wireless multi-layered sensor that allows electromyography (EMG), mechanomyography (MMG) and near-infrared spectroscopy (NIRS) measurements to be carried out simultaneously is presented. The multi-layered sensor comprises a thin silver electrode, transparent piezo-film and photosensor. EMG and MMG measurements are performed using the electrode and piezo-film, respectively. NIRS measurements are performed using the photosensor. Muscular activity is then analyzed in detail using the three types of data obtained. In experiments, the EMG, MMG and NIRS signals were measured for isometric ramp contraction at the forearm and cycling exercise of the lateral vastus muscle with stepped increments of the load using the layered sensor. The results showed that it was possible to perform simultaneous EMG, MMG and NIRS measurements at a local position using the proposed sensor. It is suggested that the proposed sensor has the potential to evaluate muscular activity during exercise, although the detection of the anaerobic threshold has not been clearly addressed.

## 1. Introduction

There is a need to develop a system for monitoring muscular activity and estimating muscular fatigue in the fields of sports science and physical rehabilitation. Many researchers have reported work in this field, and electromyography (EMG) is adopted as the primary method of evaluating muscular activity involving muscular fatigue [1,2,3,4].

EMG measures the electrical voltage induced by muscular contractions using dish electrodes that are pasted onto the surface of the skin. Additionally, mechanomyography (MMG) [5,6,7,8,9,10] and near-infrared spectroscopy (NIRS) [11,12,13,14,15] techniques have been developed for evaluating muscular activity. MMG measures the mechanical vibrations induced by muscle contraction using accelerometers, microphones, and similar equipment. NIRS measures the variations in oxygen consumption induced by muscular contractions using a light-emitting diode (LED) and photodiode (PD).

Multi-modal evaluation techniques based on two types of sensing methods, including EMG and MMG [16,17,18,19] and EMG and NIRS [20,21,22], have been developed to provide a more detailed evaluation of muscular activity as the information acquired using each of the sensing methods is based on different phenomena induced by muscular activity. In adopting these multi-mode evaluation approaches, it is necessary to ignore the fact that the sensors disturb muscular activity as several sensors are generally used. The multi-modal compact sensor based on three types of sensing methods, such as EMG, MMG and NIRS [23], has been researched. Moreover, wireless, compact or portable multi-modal sensors, such as those for EMG/MMG [24,25], EMG/NIRS [26,27], and EMG, MMG and electrical impedance myography [28], have been developed for the evaluation of muscular activity analysis.

The purpose of our research is the development of a portable wireless sensor that allows the evaluation of muscular activity and involves the prediction of muscular fatigue, i.e., the detection of the anaerobic threshold (AT). A new multi-layered sensor that allows the measurement of EMG, MMG and NIRS signals simultaneously at the same muscular motor unit has been proposed [29]. The use of the proposed sensors with an LED or PD reduces the restriction of muscle activity because three types of information are obtained by only a pair of sensors. Additionally, there are the merits that the arrangement of the sensor distance in EMG and NIRS is relatively free and that convenient multipoint sensing is allowed by the combination of one proposed sensor with an LED and multiple proposed sensors with a PD. The EMG, MMG and NIRS signals for isometric ramp contraction at the forearm have been measured using the proposed sensor, and the usefulness of the proposed sensor has been evaluated [30]. However, the measurement of muscular activity was limited because of the wired sensor. In the present work, a wireless multi-layered sensor is developed, and an experiment involving isometric ramp contraction at the forearm is conducted to evaluate whether the sensor can make simultaneous EMG, MMG and NIRS measurements. An experiment involving the cycling exercise of the lateral vastus muscle is conducted to evaluate whether it is possible to measure the local muscular activity using the sensor. Additionally, an extremal value estimated by the sensor is compared with the AT obtained by a commercial respiration gas sensor to investigate the possibility of predicting muscular fatigue using the sensor.

## 2. Methods

### 2.1. Sensing System

Figure 1 shows schematic diagrams and photographs of the multi-layered sensor. In the sensor, a polyvinylidene fluoride (PVDF) film with transparent electrodes, a transparent polyethylene naphthalate (PEN) film acting as the insulator, and a thin silver film having a hole with a diameter of 5 mm are sequentially pasted on the surface of a base plate that also has a hole with a diameter of 5 mm. Three LEDs (SMT770/805/870-40B 59-1, Epitex Inc., Japan) or a PD (HP601, Kodenshi Corp., Kyoto, Japan) are mounted on the hole in the base plate facing the films. Surface mount package types of LED and PD are selected for the miniaturization of the sensor. A current–voltage (I–V) converter circuit used for MMG is arranged on the bottom side of the base plate. Figure 2 shows a schematic diagram of the sensing by a pair of sensors with an LED or PD. The EMG is conducted using a pair of silver films and a reference electrode. The MMG is conducted using PVDF film.

### 2.2. Signal Processing Method

Figure 3 shows a photograph of the wireless multi-layered sensor (with dimensions of 100 mm × 100 mm × 55 mm) and a schematic diagram of the signal processing circuit. A portable sensor that allows wireless EMG, MMG and NIRS to be performed was developed. Figure 3b shows that the EMG signal is passed through a differential amplification circuit for noise rejection and amplification and then processed using a bandpass filter (15–500 Hz) and a notch filter. The MMG signals are processed by a bandpass filter (5–100 Hz) and notch filter. The NIRS signal is processed using a low-pass filter (25 Hz).

A micro-computer (Nucleo Board STM32F401RE; STMicroelectronics, Switzerland) having a 12-bit, five-channel analog-to-digital converter for data acquisition and control was used. ZigBee modules (XBee 802.15.4 S1; Digi International Inc, America) were used for wireless communication. In this system, two types of interrupt handlers were used for voltage measurement and transmission. Voltages in EMG, MMG (LED and PD) and NIRS were sequentially measured through interrupt processing at intervals of 100 μs. Data were thus acquired at intervals of 400 μs (2.5 kS/s). The averages of absolute values in EMG and MMG (LED and PD) derived using (1) and the averages of absolute values in NIRS derived using (2) were sequentially transmitted to a personal computer (PC) through interrupt processing at intervals of 40 ms and a serial baud rate of 38,400.
(1)$$V=\frac{1}{N}\overset{N}{\underset{n_{j}=1}{\sum }}\left|V_{n_{j}}\right|\left(j=1-3\right)$$
(2)$$V=\frac{1}{50}\overset{N}{\underset{n_{4}=351}{\sum }}\left|V_{n_{4}}\right|$$
Here $V_{n_{j}}$ is the measured voltage and *N* (400) is the number of measured voltages. In NIRS, three types of LED were sequentially switched, and (2) was used because of the measurement error generated by the switching. Data were thus obtained at intervals of 160 ms (6.25 S/s).

The Hb and HbO_2_ concentrations are derived from (3).
(3)$$\log\frac{I\left(\lambda \right)}{I_{0}\left(\lambda \right)}=\left(\varepsilon_{\mathrm{oxy}}\Delta C_{\mathrm{oxy}}+\varepsilon_{\mathrm{deoxy}}\Delta C_{\mathrm{deoxy}}\right)+S$$
Here *I*(λ) and *I*_0_(λ) are, respectively, the quantities of detected and emitted light at a wavelength of λ. *ε*_oxy_ and *ε*_deoxy_ are, respectively, the absorption coefficients of HbO_2_ and Hb [9] and *C*_oxy_ and *C*_deoxy_ are, respectively, the concentrations of HbO_2_ and Hb. Additionally, *d* and *S* are, respectively, the average light path length and the dispersion. In this paper, the unknown value of *d* is calculated to be 12.5 with reference to the literature [11]. Using the system, the concentrations of Hb and HbO_2_ were obtained at intervals of 480 ms.

The interval for transmission was optimized in a preliminary experiment. The transmission error was greater when the transmission interval was shorter. The voltage source was a 7.4-V lithium battery (LI-7100SP; 1100 mAh, 42 g, S.T.L.JAPAN, Japan), and a voltage of 3.3 V was produced for wireless communication and a voltage of 5.0 V for circuit operation. The voltage stability of the sensor system was measured after the full charging of the battery in the preliminary experiment. As a result, for this system, it was found that the voltage of 5.0 V was stable with fluctuation of ±0.05% and attenuation of 0.2% over a period of 8 h. The overall mass of the signal processing circuit in Figure 3a, including the battery, was 240 g. In the following experiments, the measurement was started by irradiating the sensor to saturate the voltage of the NIRS measurement.

## 3. Experiments

Three healthy male participants, two in their twenties and one in his forties, participated in the experiments. The experiments were conducted according to the Declaration of Helsinki. All experimental procedures were approved by an ethics committee at Saga University. The procedures and risks were explained to each participant and informed written consent was obtained from all participants. Two experiments were conducted, one involving isometric ramp contraction at the forearm and the other cycling exercise of the lateral vastus muscle.

Figure 4a shows a schematic diagram of the experimental setup and a flow chart of the isometric ramp contraction at the forearm. The sensor with the LED was pasted on the participant’s skin at a distance of 70 mm from the inside bend of the elbow, whereas the sensor with the PD was pasted in the direction of the wrist with a center-to-center distance of 25 mm between the sensors. The sensors were fixed using a transparent conductive gel. The reference dish electrode was pasted to the bony surface of the participant’s elbow for EMG. The maximum weight loaded onto the palm at which the participant could maintain a straight wrist for 3 s was measured as the maximum voluntary contraction (100% MVC) force. The participant sat on a chair, and his forearm from the elbow to the wrist was placed on a table so that the angle between the upper arm and forearm was 90°.

In the experiment, the relaxed condition of the forearm was first maintained for 30 s (Rest). Next, the condition that a vessel with a load of 400 g was placed on the participant’s palm was maintained for 10 s (Work). The isometric ramp contraction was then performed until 60% MVC was reached. The load on the palm was changed, flowing water into the vessel at a constant rate of loading. Finally, the rest condition was maintained for 30 s. EMG, MMG and NIRS signals were measured using the layered sensor.

Figure 4b is a photograph of the experimental setup and the experimental method for the cycling exercise. The pair of sensors used in the previous experiment was arranged on the belly of the left lateral vastus muscle. The reference dish electrode was pasted to the surface of the participant’s left ankle for EMG. The participant then rode on a cycling ergometer. An aero monitor (AE-301S, Minato Medical Science Co. Ltd., Japan) was arranged on the mouth of the participant to measure concentrations of O_2_ and CO_2_ in the respiration gases. The AT was estimated using the V-slope method based on concentrations of O_2_ and CO_2_ [31] and compared with that estimated using the proposed sensor [21].

In the experiment, the relaxed condition in which both feet were placed on the pedals and kept at the same height with the left leg ahead was first maintained for 1 min (rest). Next, the pedals were pushed at a speed of 60 rpm while a work load was added from 15 Watts (W) in increments of 15 W every 1 min (cycling). When the participant could no longer maintain the cycling speed, the cycling exercise was deemed complete, and, finally, the rest condition was maintained for 5 min (rest). During the experiment, EMG, MMG and NIRS signals were measured using the proposed sensor. Additionally, respiration gases such as O_2_ and CO_2_ were measured by the aero monitor. In the cycling exercise, a speed of 60 rpm was maintained through the use of a metronome.

The first experiment was conducted three times for each participant, and the second experiment once for each participant. The temperature in the room where both experiments were performed was controlled at 25 °C.

## 4. Results and Discussion

### 4.1. EMG, MMG and NIRS in Isometric Ramp Contraction

Figure 5 shows the results of three experimental runs of the isometric ramp contraction at the forearm of a participant. Figure 5a reveals that the measured EMG voltage increased with the load, whereas it was almost zero in the rest and work phases (light load). Figure 5b,c show that the MMG voltage increased slightly in the work stage and was higher in the ramp stage. It is seen that the MMG voltage in the ramp stage had a trend different from that of the EMG voltage in the ramp stage and had local maximum values, despite the measurement accuracy being insufficient to examine the local maxima. Figure 5d shows that ΔHbO_2_ and ΔHb decreased and increased during the ramp stage. These quantities then increased and decreased in the rest period after the ramp stage. It is seen that ΔHbO_2_ and ΔHb changed because the condition of the blood changed, even though there were no corresponding changes in the EMG and MMG signals. The results were in agreement with the results of muscular activity previously reported [6,14,20,24] and results obtained using a previously designed sensor with a fixed cable [30]. The simultaneous measurement of EMG, MMG and NIRS signals is thus possible using the wireless multi-sensor.

Figure 6 shows the experimental results for the three participants. The data are average values with standard deviations derived from measurements made over a period of 5 s three times per participant with spacings of 5 s. In the EMG and MMG, data were normalized by the maximum of the average value per participant. In Figure 6a–d, the results of the three participants were similar to those in Figure 5. Figure 6b,c shows that the slopes of the MMG in the ramp stage were different from those of the EMG, with the inflection points (indicated by black, blue and red dotted lines in Figure 6b,c) appearing between 75 and 84 s in Figure 6b and between 80 and 92 s in Figure 6c.

Figure 7 shows the ratio of ΔHb to the EMG voltage and the approximated quintic curve based on data between 60 and 100 s (black, blue and red lines) for detection of the AT following the literature [18]. The ratio was derived using ΔHb and EMG, normalized by the maximum and minimum average values per participant. The ratio of ΔHb to the EMG voltage had an extremal value (black, blue and red dotted lines) between 72 and 83 s. It is supposed that these are the estimated AT [21], although the AT is not actually obtained by measuring the lactic acid and the respiration gas. Additionally, the time ranges of 75 to 86 s in Figure 6b and 80 to 90 s in Figure 6c at which the inflection points in the MMG signals appeared are similar to those of the extremal values, although the measurement accuracy is insufficient to determine the inflection points precisely. It is expected that the inflection points in the MMG are related to the AT. The proposed sensor thus has the potential to detect the AT more clearly through the EMG, MMG and NIRS measurements.

### 4.2. EMG, MMG and NIRS in Cycling

Figure 8 presents the experimental results for the lateral vastus muscle of one participant in his twenties in the cycling exercise. The trends for the other two participants were similar. Figure 8a shows that the EMG voltage increased with step increments of the load. It is seen that the slope of the voltage increased slightly after approximately 7–9 min of exercise. It is considered that this increase relates to AT [21]. Figure 8b,c show that the MMG voltage increased sharply under the first load of 15 W and slightly increased or decreased with subsequent steps in the load. There was a (slightly) extreme voltage of the sensor with PD at approximately 8 min (120 W) in contrast with the case of the EMG voltage. Figure 8d shows a slight increase in ΔHbO_2_ and a reduction in ΔHb under the load of 15 W. ΔHbO_2_ and ΔHb, respectively, decreased and increased with step increments of the load from 2 min (30 W). ΔHbO_2_ and ΔHb increased and decreased during the rest period after the cycling exercise, whereas the EMG and MMG voltages during the rest period were both zero. Both the slopes of ΔHbO_2_ and ΔHb slightly decreased at around 9 min. These results relate to the AT and are similar to the change in the slope of the EMG voltage [21]. It is thus possible to make simultaneous EMG, MMG and NIRS measurements for the cycling exercise using the wireless multi-sensor. The proposed sensor offers the potential to perform muscular analysis in greater detail because the results of the different measurements have different characteristics. There were periodical oscillations at an interval of 1 s in the NIRS results owing to the muscle contraction and extension during cycling.

Figure 9 shows the ratio of Δ*Hb* to *V*_EMG_ and the approximated quintic curve based on data between 2 and 14 min (red line) to enable the detection of the threshold for the evaluation of fatigue (AT) based on the literature [21] and the relationship between oxygen uptake (VO2) and carbon dioxide output (VCO2) in the respiration gases measured by an aero monitor. In Figure 9b, the AT is obtained as the intersection of the linear regression using data after the start of the experiment (blue dotted line and dots) and using data before the end of the experiment (red dotted line and dots) using the V-slope method. The ratio of Δ*Hb* to *V*_EMG_ had an extremal value (red dotted line) at 7 min 43 s. Meanwhile, the intersection point of the slopes of VO2 and VCO2 was at 7 min 54 s. The timing of the AT obtained from the ratio of Δ*Hb* to *V*_EMG_ was thus similar to that obtained from VO2 and VCO2.

Table 1 compares the timing of the AT obtained from the ratio of Δ*Hb* to *V*_EMG_ and that obtained from VO2 and VCO2 for the three participants. It is seen that the ratio of Δ*Hb* to *V*_EMG_ can be used for determining the timing of the AT because the times are similar to those determined using VO2 and VCO2. The proposed sensor thus offers the potential to estimate the AT.

## 5. Conclusions

A wireless multi-layered sensor was developed, and simultaneous EMG, MMG and NIRS measurements were performed for an isometric ramp contraction at the participant’s forearm and cycling exercise of the lateral vastus muscle with step increments in the load. The results of the isometric ramp and cycling exercises showed that simultaneous EMG, MMG and NIRS measurements of local muscular activity were possible using the proposed sensor. Additionally, a comparison of the extremal value (the ratio of ΔHb to EMG) obtained using the proposed sensor and AT obtained using a commercial respiration gas sensor showed that the proposed sensor could be used to predict muscular fatigue.

In future work, the relationship between the extremal value, the inflection point of the MMG voltage obtained by the proposed sensor, and the AT obtained by a commercial sensor involved in the prediction of muscular fatigue will be derived for several participants and different exercises. This work will include measurements for large step increments of the load and constant loads. In the present study, the mean power frequency could not be calculated in frequency analysis because the data transmission interval was insufficient even though the sampling frequency was sufficient. Therefore, an improved system with a data logger or combined data transmission that allows frequency analysis will be developed and evaluated as one of the parameters. Additionally, the processing circuit will be smaller because it is possible to halve its size by using a smaller microcomputer and improving the circuit configuration and elements.

## Figures and Tables

**Figure 1 sensors-23-01539-f001:**
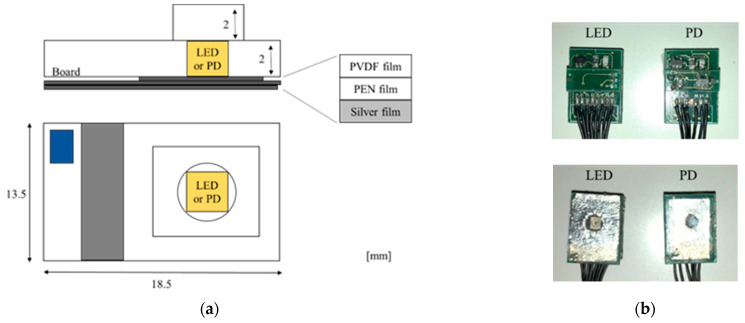
Multi-layered sensor: (**a**) schematic diagrams of the sensor and (**b**) photographs of sensors.

**Figure 2 sensors-23-01539-f002:**
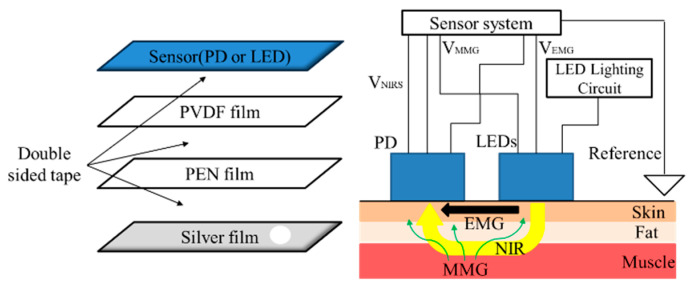
Schematic diagram of the measurement system for EMG, MMG and NIRS and changes in concentrations of oxyhemoglobin (HbO_2_) and deoxyhemoglobin (Hb) are obtained through NIRS using LED and PD, respectively. EMG, MMG and NIRS are thus measured simultaneously by a pair of sensors.

**Figure 3 sensors-23-01539-f003:**
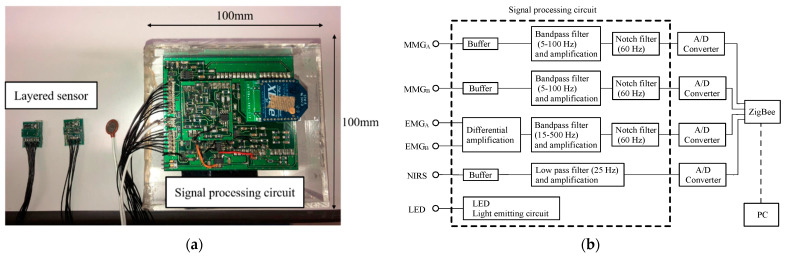
Wireless multi-layered sensor: (**a**) photograph of the device and (**b**) schematic diagram of the signal processing system.

**Figure 4 sensors-23-01539-f004:**
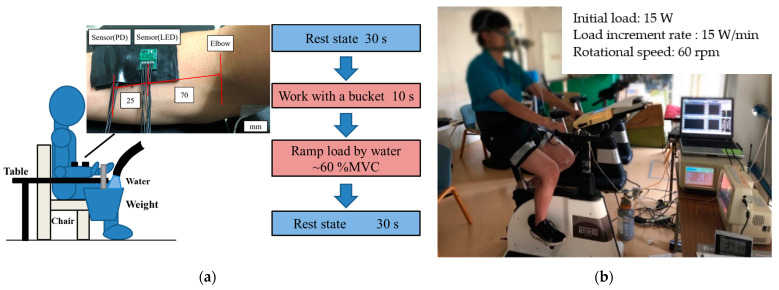
Schematic diagram of the experimental method: (**a**) isometric ramp contraction at the forearm and (**b**) cycling exercise of the lateral vastus muscle with step increments of the load.

**Figure 5 sensors-23-01539-f005:**
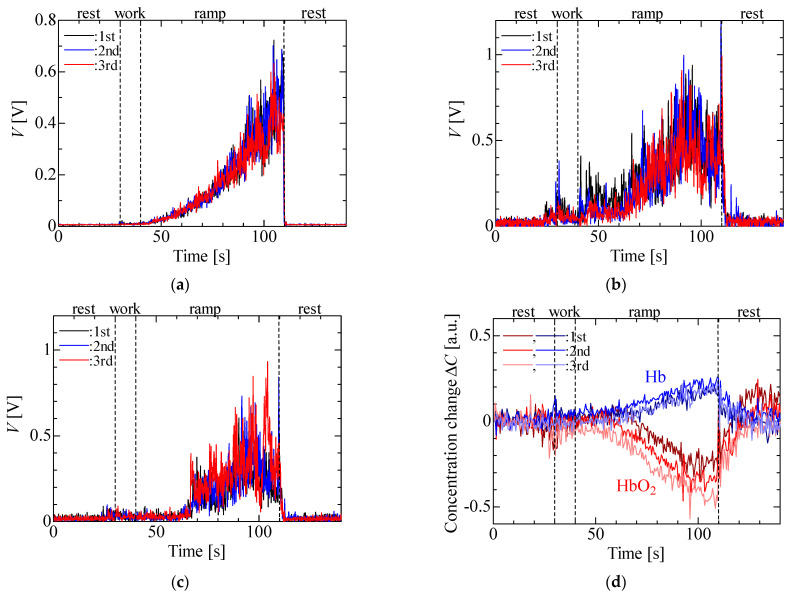
Experimental results for isometric ramp contraction: (**a**) EMG, (**b**) MMG (LED), (**c**) MMG (PD) and (**d**) NIRS.

**Figure 6 sensors-23-01539-f006:**
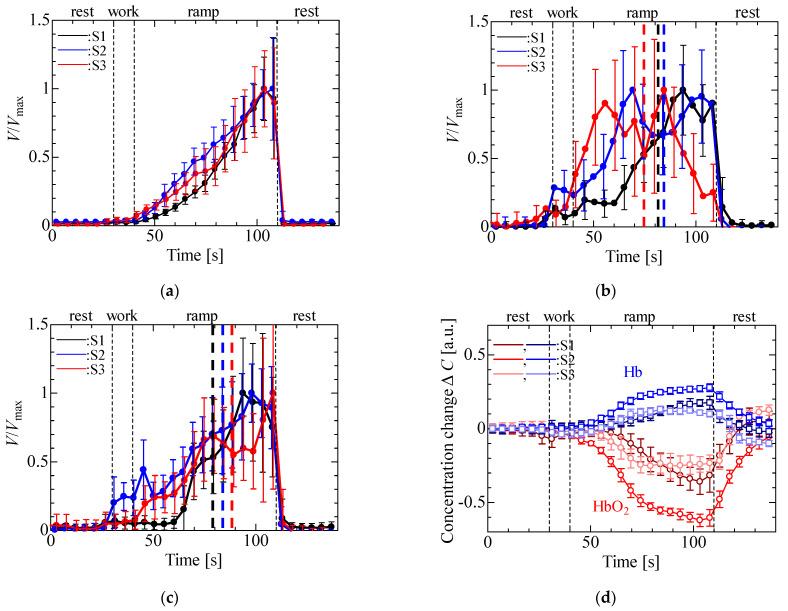
Experimental results for isometric ramp contraction: (**a**) EMG, (**b**) MMG (LED), (**c**) MMG (PD) and (**d**) NIRS.

**Figure 7 sensors-23-01539-f007:**
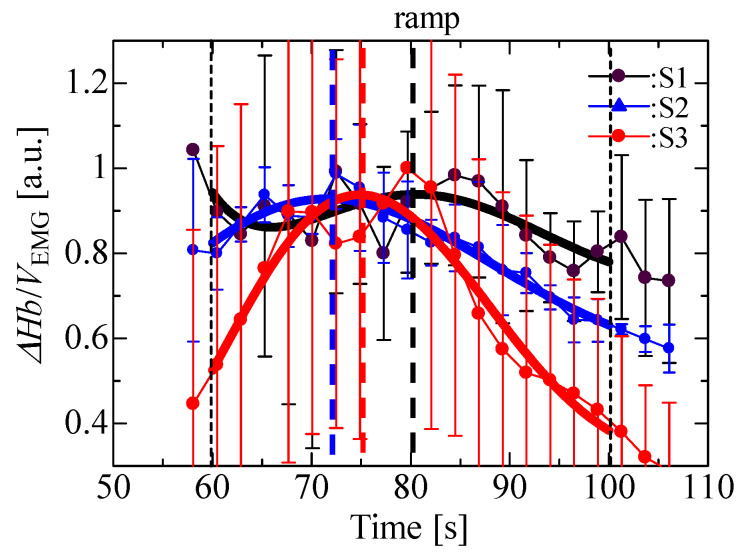
ΔHb/EMG in isometric ramp contraction.

**Figure 8 sensors-23-01539-f008:**
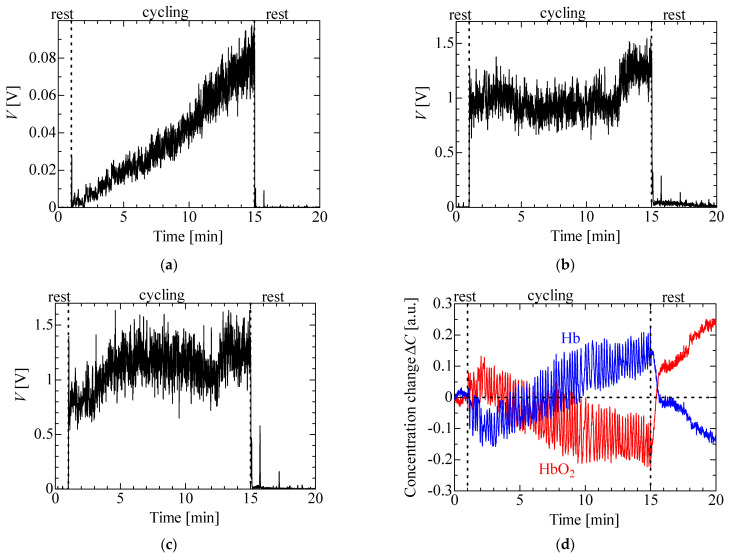
Experimental results for isometric ramp contraction: (**a**) EMG, (**b**) MMG (LED), (**c**) MMG (PD) and (**d**) NIRS.

**Figure 9 sensors-23-01539-f009:**
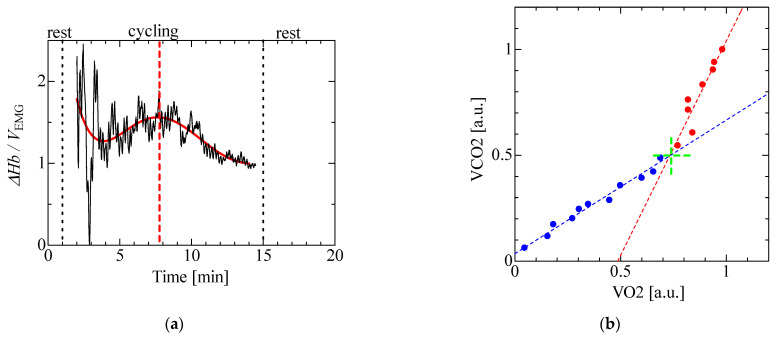
Comparison of the timings of the AT obtained from Δ*Hb*/EMG and relationship between VO2 and VCO2 in cycling exercise: (**a**) Δ*Hb*/*V*_EMG_ and (**b**) relationship between VO2 and VCO2.

**Table 1 sensors-23-01539-t001:** Comparison of the timings of the AT obtained using the proposed sensor and a commercial sensor for the three participants.

Participant	Estimated AT in the Proposed Sensor	AT in Commercial Sensor
S1	7′43	7′54
S2	7′17	7′39
S3	6′35	6′27

## Data Availability

The data presented in this study are available on request from the corresponding author.

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
