# Peer review of "A Wireless Multi-Layered EMG/MMG/NIRS Sensor for Muscular Activity Evaluation"

_sensors, 2023, doi:10.3390/s23031539_

Round 1

Reviewer 1 Report

Without doubt, the information gathered with the present study will attract a great deal of research interest. Without doubt, the work performed and described in the present manuscript brings a good amount of innovation and improvement in this field. However, I came across some parts that need revision (more explanations and/or integrations). Details of them are listed below, providing indications of the corresponding sections and lines along the manuscript.

Introduction

Line 42: Please, provide appropriated extended form for EIM acronym.

Materials and methods

Line 67: “hole with a diameter of 5 mm”: is it always the same of that mentioned in the previous row?

Line 68: Why both the three-LED (light-emitting photodiode) and the PD (photodiode) versions were considered? Authors should better explain this important point.

Lines 73-74: Please, provide the extended form of HbO2 and Hb.

Line 127: How authors considered enough (from a statistical viewpoint) performing these test by just taking into account three subjects?

Line 158: Something seems going wrong with this sentence. Please, have a check.

Line 159: Please, add appropriate reference(s) for this "V-Slope method".

Line 162: Here, the term "W" is mentioned for the first time. What does it stand for? Please, say it.

Line 169: And how many times the cycling experiment described and depicted in Figure 4(b) was performed? Three times as the isometric ramp contraction at the forearm?

Section 3 is yet Materials while 5 is results and discussion: Section 4 is missing!

Results and discussions

Figure 5: What does the acronym ΔC (expressing the ordinate of the NIRS plot) stand for? Which parameter(s) was/were exactly measured by the NIRS sensor?

Line 195: it is extremely not clear how the error was calculated for all the experiments: which were the considered references/standards?

Statements and deductions discussed from line 207 to line 212 are not clear. Please, add some information to corroborate them.

Line 219: and what about the other two subjects? Were their trends similar or different? Which differences occurred from subjects in twenties and those in forties? Why a detailed comparison using mean values and error bars, as done for the first part of the experiment (isometric ramp contraction) was not considered/shown in this case?

Line 224: Please, replace "It found" with "It was found".

Line 254: The text inserted from line 254 to the end of the section (line 265) seems to give future proposal to complete the current work. Thus, in my opinion, this part should be moved in Conclusion and emphasized as future perspective.

Figure 9b: Some additional information about the meaning of blue and red dots (what they represent and how they were calculated) should be provided.

Table 1: Completely missing caption. Please, add it!

Conclusions

Lines 275-276: Based on which parameter(s) authors say that AT measurement accuracy was insufficient? In which section reader can gather information about the computation and the achieved values of accuracy?

Reviewer 2 Report

1.- Include more references in the introduction and clearly define the purpose of the work.

2.- The conclusions must be improved

Reviewer 3 Report

Although the introduction part introduces some methods, there is not much progress on "wearable sensor", please explain the progress on this aspect related to this article.

Some references are not new. Please choose some recent references.

Please change the caption of table 1.
